# Protopine and Allocryptopine Interactions with Plasma Proteins

**DOI:** 10.3390/ijms25105398

**Published:** 2024-05-15

**Authors:** Aleksandra Marciniak, Aleksandra Kotynia, Edward Krzyżak, Żaneta Czyżnikowska, Sylwia Zielińska, Weronika Kozłowska, Marcel Białas, Adam Matkowski, Anna Jezierska-Domaradzka

**Affiliations:** 1Department of Basic Chemical Sciences, Wroclaw Medical University, Borowska 211a, 50-556 Wrocław, Poland; aleksandra.marciniak@umw.edu.pl (A.M.); aleksandra.kotynia@umw.edu.pl (A.K.); zaneta.czyznikowska@umw.edu.pl (Ż.C.); 2Division of Pharmaceutical Biotechnology, Department of Pharmaceutical Biology and Biotechnology, Wroclaw Medical University, Borowska 211, 50-556 Wrocław, Poland; weronika.kozlowska@umw.edu.pl (W.K.); anna.jezierska-domaradzka@umw.edu.pl (A.J.-D.); 3Student Scientific Club, Division of Pharmaceutical Biotechnology, Department of Pharmaceutical Biology and Biotechnology, Wroclaw Medical University, Borowska 211, 50-556 Wrocław, Poland; marcel.bialas@student.umw.edu.pl; 4Division of Pharmaceutical Biology and Botany, Department of Pharmaceutical Biology and Biotechnology, Wroclaw Medical University, Borowska 211a, 50-556 Wrocław, Poland; pharmaceutical.biology@wp.eu

**Keywords:** protopine, allocryptopine, albumin, orosomucoid, spectroscopy

## Abstract

A comprehensive study of the interactions of human serum albumin (HSA) and α-1-acid glycoprotein (AAG) with two isoquinoline alkaloids, i.e., allocryptopine (ACP) and protopine (PP), was performed. The UV-Vis spectroscopy, molecular docking, competitive binding assays, and circular dichroism (CD) spectroscopy were used for the investigations. The results showed that ACP and PP form spontaneous and stable complexes with HSA and AAG, with ACP displaying a stronger affinity towards both proteins. Molecular docking studies revealed the preferential binding of ACP and PP to specific sites within HSA, with site 2 (IIIA) being identified as the favored location for both alkaloids. This was supported by competitive binding assays using markers specific to HSA’s drug binding sites. Similarly, for AAG, a decrease in fluorescence intensity upon addition of the alkaloids to AAG/quinaldine red (QR) complexes indicated the replacement of the marker by the alkaloids, with ACP showing a greater extent of replacement than PP. CD spectroscopy showed that the proteins’ structures remained largely unchanged, suggesting that the formation of complexes did not significantly perturb the overall spatial configuration of these macromolecules. These findings are crucial for advancing the knowledge on the natural product–protein interactions and the future design of isoquinoline alkaloid-based therapeutics.

## 1. Introduction

Isoquinoline alkaloids are plant-based natural products and are considered an active constituent in numerous important medicinal herbs mainly from the families Papaveraceae, Berberidaceae, and Amaryllidaceae [1,2]. Protopine and allocryptopine are both benzylisoquinoline alkaloids (Figure 1) known for a range of pharmacological activities, including antispasmodic, antimicrobial, and anti-inflammatory effects. Protopine is characterized by a fused quaternary structure containing a benzylisoquinoline skeleton linked to a methylenedioxyphenyl group. Protopine exhibits optical isomerism due to the presence of chiral centers. It is moderately soluble in water and more soluble in organic solvents such as chloroform, ethanol, and DMSO. Allocryptopine is structurally similar to protopine, containing a benzylisoquinoline core, but with an additional methoxy group on the aromatic ring. Like protopine, allocryptopine is optically active. The compound is also slightly soluble in water and has better solubility in organic solvents. Both compounds are stable under ordinary conditions but sensitive to prolonged exposure to light, which may cause degradation [3]. Both alkaloids absorb UV light, with characteristic peaks suitable for analytical detection. The bioavailability of these two compounds in biological systems is generally low due to their moderate solubility in water. Both alkaloids exhibit similar biological properties, although specific pharmacological responses may vary due to differences in their molecular structure. The study of the interaction of alkaloids with plasma proteins, in particular with albumin, is an important topic, often discussed in the literature [4,5,6,7,8,9,10,11]. The interest in the subject is understandable because the mechanism and strength of interaction of molecules with potential pharmaceutical use with proteins present in the human body has a significant impact on the biodistribution of these substances. It is well known that a drug administered to a patient, after absorption, binds to plasma proteins or remains in the bloodstream unbound. The unbound fraction of the drug is the active fraction, while the part of the active substance bound in a complex with a protein, e.g., albumin or orosomucoid, is a kind of storage from which subsequent doses can be released. Thus, the method of binding to plasma proteins and the stability of the resulting complex has a significant impact on the action of the drug and its side effects. As mentioned above, a large group of alkaloids has already been studied in this aspect. In particular, binding interactions of berberine, palmatine, jatrorrhizine, sanguinarine, and chelerythrine have been examined in human and bovine serum albumins, hemoglobin, and lysozyme [5]. Berberine seems to be the most extensively studied of all isoquinoline alkaloids. The interactions of this compound have also been studied with the human NAD+-dependent deacetylase SIRT5 and glycosaminoglycans [12,13].

However, there have been no reports in the literature so far describing the interaction of protopine (PP) and allocryptopine (ACP) with plasma proteins. Therefore, these two isoquinoline alkaloids are of concern to us as this study would fill in the existing gap in this matter.

Plasma proteins are a large group of macromolecules that play various functions in the human body. More than 280 different blood proteins have been identified so far [14]. For the analysis with chosen alkaloids, we selected two plasma proteins: albumin (HSA) and orosomucoid (α-1-acid glycoprotein, AAG), both of human origin. These two proteins present distinct structural and physiochemical properties. They play critical roles in transport and metabolism of various substances, including drugs [15,16,17]. HSA is found in the plasma in the highest concentration. The primary structure of HSA is a single-chain polypeptide consisting of about 585 amino acids. Its molecular weight is approximately 66.5 kDa. The secondary and tertiary structures of albumin have heart-shaped structures primarily made up of alpha-helices. It does not contain beta-sheets. The structure is divided into three homologous domains (I, II, III), each consisting of two subdomains (A and B). The quaternary structure of albumin is generally found as a single monomer in solution. HSA is quite stable under a wide range of temperatures and pH, but it can be denaturated by strong acids, bases, and excessive heat. Albumin has high binding capacity for a wide range of substances, including fatty acids, bilirubin, hormones, and drugs. This binding is typically reversible and depends on the hydrophobicity and charge of the ligand [16]. On the other hand, AGG is an acute phase protein, the concentration of which increases several times in pathological conditions. The primary structure of AGG is a single polypeptide chain composed of about 183 to 201 amino acids, with a molecular weight of approximately 37–54 kDa. It is heavily glycosylated, which accounts for about 45% of its molecular mass. The carbohydrate composition can vary significantly with disease states and inflammation. The tertiary structure of AGG has a compact, globular shape, primarily formed of beta-sheets, and a small number of alpha-helices. AGG is stable at physiological pH and temperature but sensitive to glycosidase activity, which can alter its structure and function. This protein is known for its selective binding to basic and neutral drugs and has fewer binding sites than albumin but with higher specificity [15,17,18,19]. 

Due to the structural and physicochemical characteristics of these proteins, the analysis of the interaction with both of them is essential in pharmacokinetic and pharmacodynamic studies of drug candidates among the huge number of structurally diverse natural products.

Characterizing protein binding interactions is crucial in understanding biological functions and developing pharmaceuticals. Techniques such as chromatography, isothermal titration calorimetry (ITC), and spectroscopy offer different insights into these interactions. In particular, the affinity chromatography can be used to isolate and study protein–ligand interactions based on the affinity of the protein for a specific ligand, while the size-exclusion chromatography (SEC) is used to monitor changes in molecular size due to complex formation [20,21]. The ITC measures the heat change that occurs during the binding of a ligand to a protein, which is indicative of the binding affinity and thermodynamics [22]. Both methods have their limitations in terms of studying protein–ligand interactions. The chromatographic methods require relatively large amounts of protein and may not always reflect physiological condition if the matrix affects the protein structure or function or if nonspecific binding to the column material occurs [23]. In the case of ITC, although it provides a complete thermodynamic profile of the interaction, the method requires relatively large amounts of sample compared to spectroscopic techniques. Moreover, it requires meticulous sample preparation to avoid heat effects not related to binding. Also, the sensitivity might be insufficient for very weak or very strong interactions [22,24].

Due to the fact that proteins, under the influence of interaction and binding into a complex with small molecules, can change their structure, spectroscopic methods are considered most suitable for investigating these types of changes. Spectroscopy measures the interaction of matter with electromagnetic radiation and can be utilized to study protein binding by observing changes in spectral properties upon ligand binding. This highly sensitive method provides information about changes in secondary and tertiary structures of proteins and requires small sample quantities. Moreover, fluorescent and circular dichroism spectroscopy methods can deliver significant insights because protein molecules contain fluorescent amino acids and chiral carbon atoms. Thus, using fluorescence spectroscopy, it is possible to monitor changes in fluorescence intensity or emission wavelength, useful for studying changes in protein’s tertiary structure or the environment of fluorescent residues. In turn, circular dichroism is used to analyze changes in secondary structure of protein upon ligand binding. Furthermore, the UV-Vis spectroscopy detects shifts in UV absorption, thus supplementing the analytical part with the necessary information on the changes in protein conformation or environment upon ligand binding [25,26,27,28]. 

For the reasons described above, we used the following three methods: fluorescence spectroscopy, circular dichroism, and UV-Vis spectroscopy. In addition, the spectroscopic studies were supported by computational methods, i.e., molecular docking and molecular dynamics simulation.

## 2. Results

### 2.1. Determination of Binding Constants

Previous studies have shown that protopine and allocryptopine exhibit fluorescence at an excitation wavelength of 285 nm, which is also commonly used to excite protein fluorescence [29]. Due to the overlapping effect in fluorescence characteristics, it has been challenging to determine the nature of the interactions between PP and ACP and plasma proteins using fluorescence spectroscopy alone. This might explain a deficit of literature describing these interactions. Hence, we decided to use UV-Vis spectroscopy to examine these interactions. The blood plasma proteins are multifunctional macromolecules which share structural similarities, resulting from the fact that they are composed mainly of L-amino acids. Consequently, characteristic, typical peaks for these molecules exist in the UV absorption spectra. Proteins exhibit bands at 180–230 nm and 260–280 nm wavelength range due to π–π* energy transfer which is exhibited by peptide bonds and aromatic side chains of amino acids, respectively [30,31]. The titration of protein solution (HSA and AAG) by studied alkaloids (ACP and PP) changed both the absorbance and the position of the maximum peak. This methodological pivot allowed us to explore and elucidate these interactions more effectively.

Generally, the addition of the investigated alkaloids to the protein solution affected an increase in absorbance (Figure 2). These results demonstrated the hyperchromic effect. The value of the percentage of chromaticity is a little bit larger for PP (71.50%) than for ACP (65.52%) in the interaction with HSA. A similar trend was shown for the interaction of these alkaloids with AAG, where PP caused 81.93% hyperchromicity while ACP showed 73.50% (Table 1). Also, the 2–3 nm of red shift occurred after alkaloids addition. These results confirm peptide/alkaloid complex formation, for which the equilibrium reaction can be expressed as follows (Equation (1)):(1)$$protein+alkaloid\overset{K_{app}}{↔}protein/alkaloid\;complex$$

The equilibrium of apparent association constants (K_app_) presents the following equation (Equation (2)):(2)$$K_{app}=\frac{\left[complex\right]}{\left[protein\right][alkaloid]}$$

The apparent association constants (K_app_) describing the interaction between components can be determined by Benesi–Hildebrand procedure via monitoring the changes in UV absorption spectra [32]. The linear plot of double-reciprocal function (Figure 3A,B) suggests that both studied alkaloids are binding with HSA and AAG with a similar impact. The complex formation processes are spontaneous due to the negative value of standard Gibbs free energy (ΔG) of these equilibrium reactions (Table 1). The comparison of the apparent association constant (K_app_) values suggests that the alkaloids exhibit more stable complexes with HSA than with AAG (Table 1). Furthermore, ACP shows a stronger affinity for both proteins than PP and a higher affinity to HSA than to AAG. It can be concluded that the ACP is bound with HSA with the highest affinity. Additionally, the stoichiometry analysis of the complexes’ stoichiometry was performed. The Scatcher plots for all studied systems are linear functions and suggest the formation of equimolar complexes (Figure 3C,D). The calculated number of binding sites (n) in proteins is shown in Table 1, and all the values approach one.

### 2.2. Analysis of Binding Sites and Interaction Modes

The results obtained from spectroscopic measurements indicate the interaction of ACP and PP with HSA and AAG and the complex formation in all analyzed systems. Molecular docking studies and molecular dynamic simulation were used to characterize the mode of interactions and predict the binding site. First, a molecular docking procedure for two Sudlow’s drug binding sites of human albumin, site 1 (IIA) and site 2 (IIIA) [33,34], was performed. The binding affinity for the best conformers is shown in Table 2. All values are negative, which indicates the possibility of the formation of stable complexes. For allocryptopine, it was −6.3 kcal/mol for site IIA and −7.7 kcal/mol for site IIIA. For protopine, it was −6.5 kcal/mol and −7.7 kcal/mol, respectively. These results suggest that site IIIA is slightly more preferable. In the second step, the stability of the complexes and the binding free enthalpy were calculated using the molecular dynamic method. The conformer with the most negative binding affinity, obtained from the docking study, was simulated for a 100 ns period. As shown in Figure 4, for the HSA-ACP system, where allocryptopine is docked at site 1, the complex stabilizes after 60 ns. After stabilization, the averaged root mean square deviation (RMSD), structural drift from the initial position, was calculated as 0.97 ± 0.04 nm. For the HSA-ACP complex, where the allocryptopine is docked at site 2, the structural fluctuations are smaller. After about 2 ns, the complex is stable. The averaged RMSD value from 2 to 100 ns was calculated as 0.31 ± 0.03 nm. Also, for the HSA-PP system, smaller fluctuations were observed for the protopine docked at site 2. After about 10 ns, complex was stable, and the averaged RMSD (10–100 ns) was 0.25 ± 0.02 nm. For the ligand docked at site 1, the complex stabilizes after about 55 ns, and the average RMSD is 0.49 ± 0.04 nm. The compactness of the HSA is characterized by the radius of gyration (Rg). The plot of Rg values against simulation time is presented in Figure 5. At the starting point, the Rg value of the unliganded HSA is 2.69 nm. During the simulation, only slight fluctuations were observed. For complexes with docked ligands in site 2, the initial Rg value is close to unliganded HSA, i.e., 2.68 nm for ACP and 2.69 nm for PP. After simulation time, for both complexes, Rg is slightly lower than for free albumin (Figure 5). This means that HSA is slightly more compact after binding allocryptopine and protopine. For the systems with ligands docked at site 1, the initial Rg and that after 100 ns are larger than those for site 2 (Figure 5). In the third step, the binding free energy was calculated from MD complex trajectories using the gmx_MMPBSA tool [35]. The calculation was performed with frames after complex stabilization. The results are given in Table 2. The average ΔG_bind_ was found to be −17.6 kcal/mol and −20.1 kcal/mol for the HSA-ACP system with ligands at site 1 and site 2. For the HSA-PP complex, it was found to be −12.3 kcal/mol and −18.4 kcal/mol, respectively. The ΔGbind values are smaller for site 1 than for site 2. In summary, the in silico studies quite clearly suggest that drug binding site 2 (IIIA) should be preferred over site 1 (IIA). In the next part of this work, we will confirm this hypothesis using binding site markers.

These findings were verified by experimental methods. We conducted competitive studies with markers that bind to specific sites in protein molecules. This type of research allows us to prove that a stable complex is formed between the macromolecule and the tested compound in a specific way. Fluorescence spectroscopy was used to consider this issue. The HSA is an essential factor, which plays a crucial role in the pharmacokinetics of many drugs, affecting their efficacy. Moreover, albumin is a multi-carrier agent that also transports fatty acids and endogenous and exogenous ligands. The albumin has two well-known drug binding Sudlow’s sites, i.e., site 1 and site 2, which are the most important in drug transport. Binding site 1 is specific for heterocyclic compounds, and site 2 is the preferred interaction site for aromatic compounds [34,36]. The binding displacement study can indicate which of these cavities is more favorable for ligand. The dansyl amino acids exhibit the phenomenon of fluorescence. The dansylated glycine (dGly) is a marker for drug binding site 1, while the dansyl phenylalanine (dPhe) enters binding site 2 [19,37]. The dansyl amino acids possess different physicochemical properties because of differences in side chains, and it determines which binding site in the protein is preferable. The dGly has no side chains and is specific for site 1 as good as different polar dansylated amino acids (e.g., dAsn, dGlu, dArg). However, dPhe with a hydrophobic phenyl ring is dedicated to the second drug binding site [37]. The titration of complex HSA/dGly with studied alkaloids, i.e., ACP and PP, was performed. Both alkaloids lead to an increase in fluorescence intensity with a slight movement to a higher energy wavelength (Figure 6A,B). This indicates that these compounds do not occupy the binding site 1. The observed increase in the intensity of the tested band is probably related only to the change in the concentration of reagents in the analyzed solution. However, the stepwise addition by steps of ACP and PP to the solution with HSA/dPhe induced the reduction in the fluorescence intensity (Figure 6C,D), which suggests the marker exchange by alkaloids. In the final condition, the ACP caused about 12% of dPhe replacement, whereas for PP, the obtained value is more than 6% (Table 3). Undoubtedly, drug binding site 2 is a more desirable interaction space for ACP and PP alkaloids. These findings well correspond to K_app_ values calculated from the UV-Vis spectroscopy results. As described above, the ACP has a higher affinity to HSA than the PP compound (Table 2).

Figure 7 and Figure 8 show the position of ACP and PP at albumin’s binding site 2 and a 2D plot of interaction mode. The orientation in the pocket and the interactions for both compounds are very similar. One hydrogen bond, between the nitrogen atom at Lys414 and the oxygen atom in the dioxolane ring, has been found. 1,3-Benzodioxole moiety also interacts with HSA via contacts with Leu387 (π-alkyl), Arg410 (π-cation), and Ser489 (π-donor). The middle ring of the molecule is involved in π-sigma and alkyl interactions with Asn391, Arg485, Leu407, and Leu453. The second 1,3-Benzodioxole group of protopine or dimethoxyphenyl moiety of allocryptopine is located deeper in the pocket and interacts using hydrophobic contacts, π-alkyl, and alkyl with Phe403, Leu453, Ala449, Val433, Cys392, Gly434, Cys383, Leu430, and Ile388.

The in silico studies also characterized a binding mode for interactions between ACP and PP with α1-acid glycoprotein. The binding affinity, from the molecular docking approach, was calculated as −8.8 kcal/mol for AAG-ACP and −9.8 kcal/mol for AAG-PP (Table 2). This suggests strong interactions and complex formation. The stability of the complexes was simulated for 100 ns using the molecular dynamics method. The results are shown in Figure 9. For AAG-ACP, the system stabilizes after about 10 ns. The averaged root mean square deviation (RMSD) was calculated as 0.20 ± 0.04 nm (10–100 ns). This is only slightly more than that for unbound protein (0.17 ± 0.03 nm). For AAG-PP, RMSD increases rapidly from 8.5 to 10 ns. Some fluctuations are observed between 10 and 20 ns. After 20 ns, the complex is stable. The averaged RMSD was calculated as 0.57 ± 0.04 nm (20–100 ns). The plot of Rg values against simulation time is presented in Figure 10. At the initial state, the Rg value of the free AAG, AAG-ACP, and AAG-PP was found to be 2.69 nm for all systems. During the simulation, some fluctuations were observed. After simulation time, for both complexes, Rg is slightly higher than that for free AAG (Figure 10). However, the average values are close to each other, i.e., 1.62 ± 0.01 nm, 1.61 ± 0.02 nm, and 1.61 ± 0.02 nm for AAG, AAG-ACP, and AAG-PP, respectively. The binding free energy, calculated from dynamic simulation, is given in Table 2. The average ΔG_bind_ after stabilization was found to be −17.0 kcal/mol for AAG-ACP and −23.3 kcal/mol for AAG-PP. Negative energy indicates the formation of complexes.

The competition binding assay was also used to characterize the alkaloids’ interactions with AAG. The AAG, similar to HSA, is one of the major drug transporters in plasma protein. The AAG usually binds basic molecules, but it is also able to associate neutral lipophilic and some acidic compounds [38]. In the AAG molecule, two drug binding sites can be distinguished, which are suitable for basic ligands and one for acidic molecules [39]. However, only one main site is of clinical relevance [40]. One of the most commonly used markers for examining the binding site in AGP is quinaldine red (QR) which is a specific fluorescent probe [41,42,43]. Stepwise addition of ACP and PP to the AAG/QR complex resulted in a decrease in fluorescence intensity (Figure 6E,F). This effect proved that the studied alkaloids replaced the QR in the AAG binding cavity. In the tested concentration range, the exchange of the QR marker by ACP occurs to a greater extent. The percentage of replacement was equal to 71.9% here, while PP only reached 53.7% (Table 3). This observation is consistent with the value of the interaction constant K_app_ of alkaloids with AAG, which is slightly higher for ACP (Table 1).

Figure 11 and Figure 12 show the position of allocryptopine and protopine at the α1-acid glycoprotein’s binding pocket and a 2D plot of interaction mode. The location in the cavity and the interactions for both compounds are very similar. 1,3-Benzodioxole moiety is situated deeper at the pocket and interacts with Phe49 (π-alkyl), Leu62 (alkyl), and Tyr127 (π-donor). One hydrogen bond, between the oxygen atom at Ser125 and the carbonyl group of a middle 10-membered ring, is formed. The second 1,3-Benzodioxole ring (PP) or dimethoxyphenyl moiety (ACP) is located at the entrance to the pocket. The alkyl contacts with Tyr37, π-alkyl with Val, and π-π stacked with Phe32 were found.

### 2.3. The Changes in Protein Secondary Structures

We have also used CD spectroscopy, which is a useful method to observe the protein spectra after the interaction with small molecules [44]. This analytical technique allows to see every change in the secondary or tertiary structure, which is manifested by changes in the course of the recorded spectrum. Both analyzed macromolecules, i.e., HSA and AAG, show spectral characteristic of the predominant α-helix and β-sheet structures, respectively. Two negative bands were visible in the spectrum for albumin, located near 209 and 220 nm (Figure 13). However, the dominating β-sheet structure in the case of AAG was characterized by one negative band located around 220 nm (Figure 13) [45]. The CD spectra were also measured after adding a small portion of ACP and PP (Figure 13). It was observed that, with the increase in the alkaloid concentration, the course of the spectra practically did not change. The intensity of the bands shifted slightly towards less negative values.

The obtained results were also analyzed by the CD Multivariate SSE program. This analysis made it possible to calculate the percentage of individual forms of the secondary structure in protein molecules, both before the addition and during the increase in concentration of ACP and PP in the tested systems. Calculated values are summarized in Table 4 and Table 5. The obtained values also confirm the insignificant influence of ACP and PP on the secondary structure of the tested plasma proteins. The content of the α-helix in the HSA molecule decreases by 0.9% in the case of ACP and by 0.3% in the case of PP, with a five-fold excess of the analyzed alkaloids (Table 4). In the case of AAG, this change is even smaller and amounts to 0.1% and 0.2% for ACP and PP, respectively (Table 5). Therefore, summing up, the CD results clearly show that the formation of complexes between the tested alkaloids and plasma proteins does not affect the spatial structure of the analyzed macromolecules. 

## 3. Discussion

The experimental investigation into the interactions between human serum albumin (HSA), α-1-acid glycoprotein (AAG), and the alkaloids allocryptopine (ACP) and protopine (PP) has provided valuable insights into the binding dynamics and affinities of these compounds. Both analyzed proteins have Trp residues in their sequences: one in HSA and three in AAG [19]. Therefore, fluorescence spectroscopy could potentially be a well-chosen method to determine how PP and ACP interact with analyzed macromolecules and to calculate the binding constants for the complexes formed. Fluorescent spectroscopy has been widely employed in investigating drug–protein interaction due to its sensitivity and specificity. Numerous studies have utilized this technique to elucidate the binding mechanisms and affinities of various compounds with plasma proteins [46]. Unfortunately, the spectroscopic properties of the alkaloids selected for our research stand in the way. Kubala and co-workers proved in their work that both protopine and allocryptopine exhibit the phenomenon of fluorescence at the excitation line of 285 nm [29]. Unluckily, these are the same conditions used to excite protein fluorescence. Thus, it was not possible to observe the interaction between PP, ACP, and analyzed plasma proteins with fluorescence spectroscopy because the effect for both groups of molecules overlapped. Therefore, these compounds have not yet been described in the literature in this aspect. For example, the extensive studies have been carried out on other isoquinoline alkaloids such as berberine, chelerythrine, and sanguinarine. Data revealed that the formation of berberine/HSA complexes is thermodynamically spontaneous, exothermic process with the association constant of 4.071 × 10^4^ dm^3^mol^−1^ (close to the value obtained here for ACP) [47]. Beyond berberine, chelerythrine, and sanguinarine, other isoquinoline alkaloids have also been extensively studied for their interactions with plasma proteins. For instance, studies on palmatine have shown its binding affinity with HAS and AAG, shedding light on the thermodynamics and structural changes introduced by its interaction with these proteins [48]. Our study revealed that ACP and PP have negligible impact on the secondary structure of plasma proteins. In turn, Li et al. [44] demonstrated that berberine affects the structure of protein through the reduction in the number of α-helices. The interactions of berberine in the Sudlow site I of HSA is primarily influenced by hydrophobic and electrostatic forces. Similarly, sanquinarine has been shown to interact with HSA within subdomain IIA [49]. The association constants in this case are equal to 2.18 × 10^5^ dm^3^mol^−1^ and 5.97 × 10^4^ dm^3^mol^−1^ for alkanolamine and iminium forms, respectively. The data also indicated that the most stabilizing factors are van der Waals interactions and hydrophobic forces. Interestingly, π-electrons of aromatic system of the alkanolamine form of alkaloid can interact with the positively charged binding sites of protein. The structural changes in the protein are influenced by decreasing the α-helix and increasing the random coil structure. In addition to experimental methods, computational techniques such as molecular docking and molecular dynamics simulation have been instrumental in understanding the binding modes and energetics of ligands with plasma proteins. These computational studies complement experimental findings by providing atomistic insights into the binding interactions and conformational changes induced upon ligand binding [50]. Our experimental and computational findings demonstrated a stronger preference for protopine and allocryptopine to bind to site IIIA of the protein. In contrast, chelerythrine primarily interacted with the HSA at the Sudlow I site [51]. The association constants for both charged iminium and neutral alkanolamine forms, respectively, are equal to 1.22 × 10^6^ dm^3^mol^−1^ and 1.87 × 10^5^ dm^3^mol^−1^. The iminium form of alkaloid primarily engages through electrostatic interactions, particularly in the exothermic binding process. Conversely, the hydrophobic forces are important in the endothermic binding of the alkanolamine form within the protein’s binding cavity. Various biophysical techniques, including X-ray crystallography, NMR spectroscopy, and mass spectrometry, have been employed to characterize the binding sites of small molecules on plasma proteins. These studies have revealed the structural determinants governing ligand binding and specificity, aiding in the rational design of novel therapeutics [52]. Understanding the binding interactions between drugs and plasma proteins is crucial for rational drug design and optimization. Insights gained from these studies can guide medicinal chemists in designing compounds with improved pharmacokinetic properties, such as enhanced plasma protein binding affinity and metabolic stability [53].

## 4. Materials and Methods

### 4.1. Materials

The powder plasma proteins, i.e., human serum albumin (HSA) (Sigma-Aldrich Ltd., St. Louis, MO, USA) and human α-1-acid glycoprotein (AAG) (Sigma-Aldrich Ltd., St. Louis, MO, USA), were used without any purification. The protein samples were prepared in phosphate-buffered saline solution at pH = 7.4 (Sigma-Aldrich Ltd., St. Louis, MO, USA), and the concentration was 1 × 10^−6^ M. The markers, i.e., dansylated-L-glycine (dGly), dansylated-L-phenylalanine (dPhe), and quinaldine red (QR), were dissolved in ethanol (Sigma-Aldrich Ltd., St. Louis, MO, USA), and stock solutions of concentration of 1 × 10^−3^ M were prepared. The stock solution of alkaloids was obtained by dissolving them in absolute ethanol (Merck KGaA, Darmstadt, Germany), and its concentration was 1 × 10^−3^ M.

### 4.2. Methods

#### 4.2.1. UV Absorption Spectroscopy

The Jasco V750 spectrophotometer (Jasco, Tokyo, Japan) was used to record the absorption spectra. The measurements were carried out in the wavelength range of 250–310 nm with a 0.5 nm interval, and the temperature was 297 K. All spectra were collected in quartz cells with a 10 mm path length. The baseline correction was performed using the phosphate-buffered solution. Each spectrum was measured using 3 accumulations, and the final result is the average of all measurements. The proteins solution of HSA or AAG were titrated with studied compounds, ACP or PP, to obtain the following molar ratios of proteins HSA/AAG to alkaloids ACP/PP: 1:0.5, 1:1, 1:1.5, 1:2, 1:2.5, 1:3, 1:3.5, 1:4, 1:4.5, 1:5, 1:6, 1:7, 1:8, 1:9, and 1:10. The changes in absorption peak were used to calculate the apparent association constants K_app_ by the Benesi–Hilderbrand Equation (3) [32]: (3)$$A_{obs}\left(1-\alpha \right)c_{0}\varepsilon_{P}l+\alpha c_{0}\varepsilon_{c}l$$
where A_obs_—protein solution absorbance with different alkaloid contents at 283 nm, α—the degree of association between protein and alkaloid, ε_p_—protein molar absorptivity, ε_c_—the complex molar absorptivity, c_0_—the initial protein concentration, and l—the optical path length. Based on Lambert–Beer law, the cεl can be replaced by absorbance A, and then, Equation (3) takes the following form (4):(4)$$A_{obs}=\left(1-\alpha \right)A_{0}+\alpha A_{c}$$
where A_0_—the protein solution absorbance, and A_c_—the protein/alkaloid complex absorbance. The absorbance changes at 285 nm (HSA) and 283 nm (AAG) as a relation of double-reciprocal concentration are a linear function that allows to determine the constant K_app_ by UV data fitting to function (5):(5)$$\frac{1}{A_{obs}-A_{0}}=\frac{1}{A_{c}-A_{0}}+\frac{1}{K_{app}\left(A_{c}-A_{0}\right)c_{a}}$$
where c_a_—the alkaloid concentration in each step. The number of binding sites n was estimated by the Scatchard Equation (6) [54]:(6)$$\frac{r}{c_{f}}=nK-Kr$$
where r—the number of alkaloid moles bound to one mole of protein, c_f_—the free concentration of compound, n—the number of equivalent binding sites by protein, and K—the binding affinity constant. The standard Gibbs free energy (ΔG) for binding equilibrium was evaluated from Equation (7):∆G = −RTlnK(7)
where R—the gas constant, 8.314 Jmol^−1^K^−1^, T—the temperature, 298 K, and K—the binding constant. 

#### 4.2.2. Fluorescence Spectroscopy

The Cary Eclipse 500 spectrophotometer (Agilent, Santa Clara, CA, USA) was used to perform fluorescence studies. The measurements were carried out at 297 K and were conducted in a quartz cuvette with a 10 mm path length. The excitation wavelength for HSA with dansylated amino acids complexes was equal to 350 nm, whereas for the AAG/QR set, it was 500 nm. The sample solution was prepared by mixing protein (HSA or AAG) with an equimolar marker (dGly or dPhe or QR) solution, and then, alkaloids (ACP or PP) were added in appropriate amounts. The percentage of exchange marker in the protein cavity was calculated by Equation (8): (8)$$PMR=\frac{F_{0}-F}{F_{0}}\cdot 100\%$$
where PMR—the percentage of marker replacement in blood plasma protein, F_0_—the steady-state fluorescence intensities of protein with marker solution, and F is the steady-state fluorescence intensities after the addition of alkaloids. Each spectrum was measured using 3 accumulations, and the final result is the average of all measurements.

#### 4.2.3. Circular Dichroism Spectroscopy

The Jasco J-1500 magnetic circular dichroism spectrometer (JASCO International CO., Tokyo, Japan) was used in circular dichroism spectroscopy. Spectra were collected at room temperature with a 10 nm path length. All HSA and AAG spectra, in the absence and presence of alkaloids, were baseline corrected. The range of scanning was 205–250 nm with a 50 nm/min scan rate speed and response time of 1 s. Three accumulations were used during all measurements, and the final result is the average of all measured spectra. Two milliliters of the protein solution was used, and a small portion of alkaloids (stock solution) was added to give the following molar ratios of PP/ACP to HSA/AAG: 0:1, 0.5:1, 1:1, 2:1, 3:1, and 5:1. The percentage of content of the secondary structure elements in proteins was analyzed by CD Multivariate SSE programs (JASCO International CO., Tokyo, Japan), with the use of mean residue molar concentration of proteins. 

#### 4.2.4. In Silico Studies

Three-dimensional ligand structures were optimized in the Gaussian 2016 C.01 software package [55]. The crystal structures of HSA (2bxg) and AAG (1kq0) were obtained from the RCSB Protein Data Bank. Ligands and proteins were prepared using AutoDock Tools 1.5.6 [56]. From the protein structure, co-crystallized molecules and water were removed. Kollman partial charges and non-polar hydrogens have been added. ACP and PP were prepared as follows: rotatable bonds were ascribed, nonpolar hydrogens were merged, and partial charges were added. AutoDockVina 1.1.2 [57] was used for docking. The grid box was set according to the binding pocket site in the crystal structure. The docking protocol was first validated by self-docking of the crystallized ligands. The visualizations were performed using the ChimeraX 1.4 software [58] and LigPlot + v.2.2 software [59]. The Gromacs 2021.2 software [60] was used for molecular dynamic simulation. CHARMM36m [61] force field was applied. The topology and the ff parameters were generated using the CHARMM-GUI server [62,63]. Molecules were solvated in a regular box of the TP water model, and the box was extended to 10.0 Å beyond the protein structure and was neutralized with KCl (0.15 M). The cut-off was set to 1.2 nm for long-range van der Waals and electrostatic interactions. After system energy minimization (5000 steps), the system was equilibrated for 125 ps. Next, MD simulation was conducted for the period of 100 ns with the step of 2 fs, considering a constant pressure of 1 bar and a constant temperature of 303 K. The binding free energy ΔG_bind_ was calculated using the gmx_MMPBSA tool v.1.6.1 [35].

## 5. Conclusions

The study effectively demonstrates that both ACP and PP can spontaneously bind to HSA and AAG, forming stable complexes, with ACP showing a higher affinity for both proteins. This novel finding contributes to our understanding of how these specific alkaloids interact with major plasma proteins.

Despite challenges with fluorescence spectroscopy due to the overlapping spectral properties between the proteins and alkaloids, the study successfully integrated different techniques such as UV-Vis spectroscopy, molecular docking, competitive binding assays, and circular dichroism (CD) spectroscopy. This multi-technique approach provided a comprehensive and nuanced view of the binding dynamics. Additionally, the in silico investigations uncovered new information by providing evidence for the preference of the site 2 (site IIIA) binding location of human serum albumin to both alkaloids, allocryptopine and protopine, compared to site 1 (site IIA). This preference was manifested through more negative docking energies, quicker and more stable complex formation during molecular dynamics simulations, and more favorable binding free energy profiles. This is a significant advancement in understanding the interaction specifics at the molecular level.

The findings have substantial implications for natural product-based drug design and delivery. By elucidating the nature of HSA-binding of these bioactive plant metabolites, the study opens new pathway for enhancing the bioavailability and efficacy of therapeutics derived from natural products. This is particularly important in the context of developing drugs with optimized delivery mechanisms based on protein binding characteristics. 

Overall, in this study, we described findings regarding behaviors of protopine and allocryptopine with key plasma proteins, utilizing a novel combination of experimental and computational approaches to provide deeper insights into the pharmacokinetic properties of natural compounds.

## Figures and Tables

**Figure 1 ijms-25-05398-f001:**
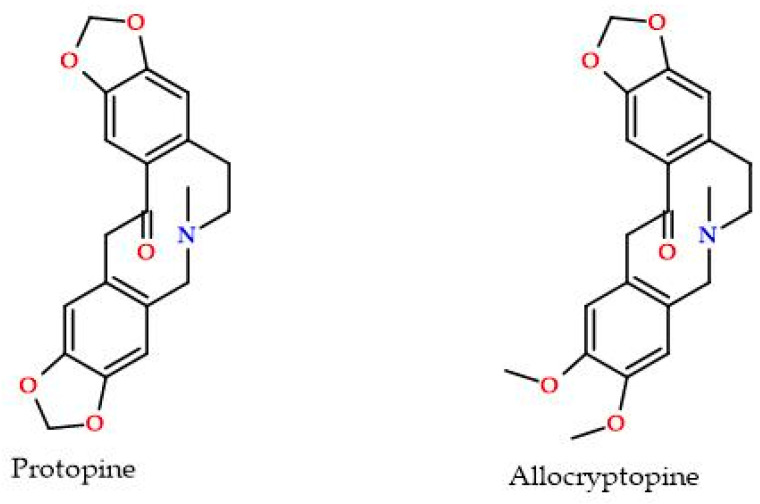
Protopine and allocryptopine molecular structures.

**Figure 2 ijms-25-05398-f002:**
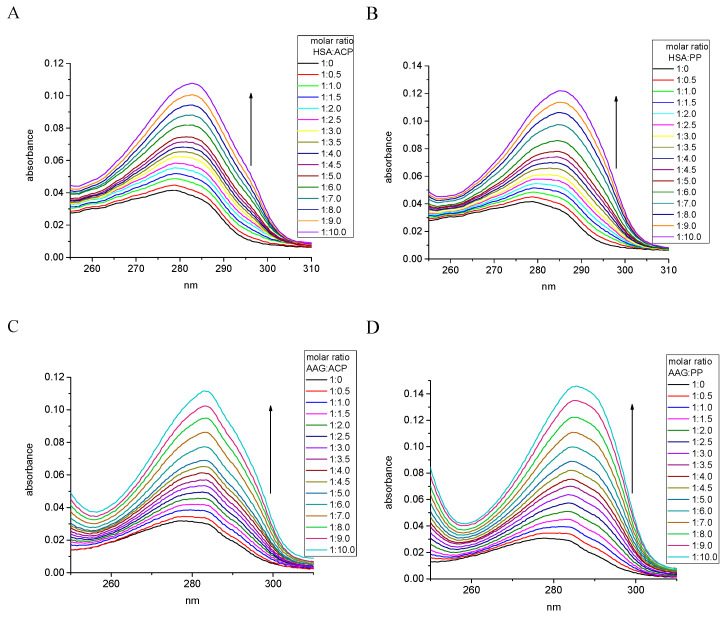
The changes in UV absorption spectra during the titration of (**A**) HSA by ACP, (**B**) HSA by PP, (**C**) AAG by ACP, and (**D**) AGG by PP.

**Figure 3 ijms-25-05398-f003:**
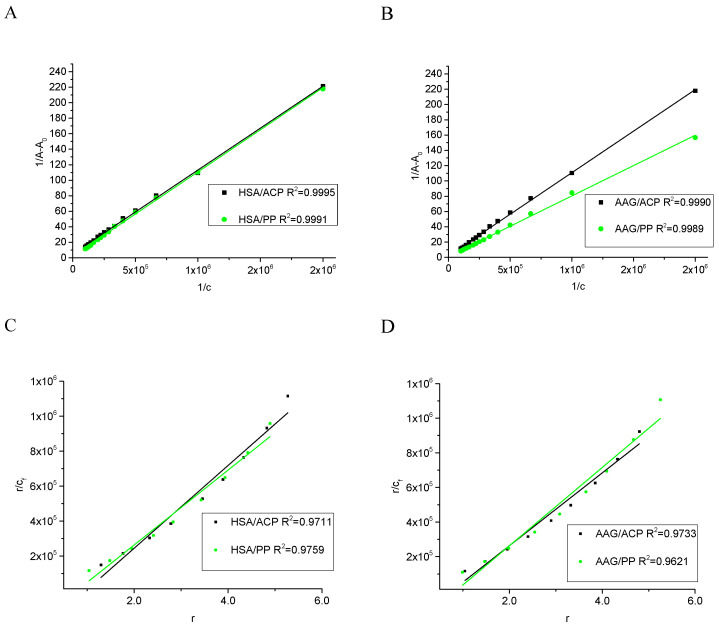
The UV absorption data analysis plot: the Benesi–Hildebrand procedure for titration of HSA or AAG by (**A**) ACP (**B**) PP and the Scatcher plot for titration of HSA or AAG by (**C**) ACP and (**D**) PP.

**Figure 4 ijms-25-05398-f004:**
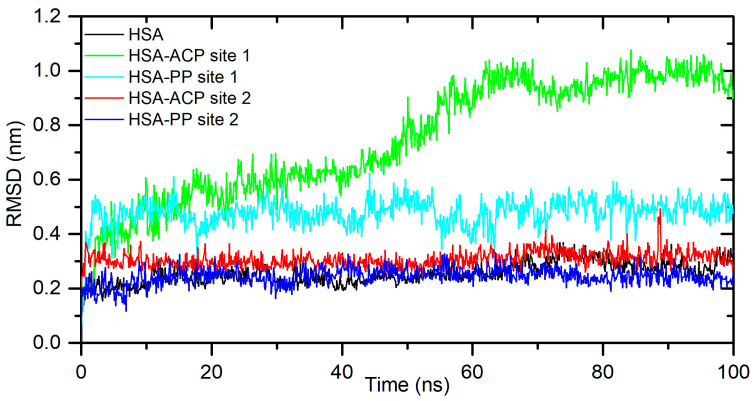
The backbone atoms’ RMSD plots during 100 ns MD simulation for unliganded HSA and with ACP and PP docked at site 1 and site 2.

**Figure 5 ijms-25-05398-f005:**
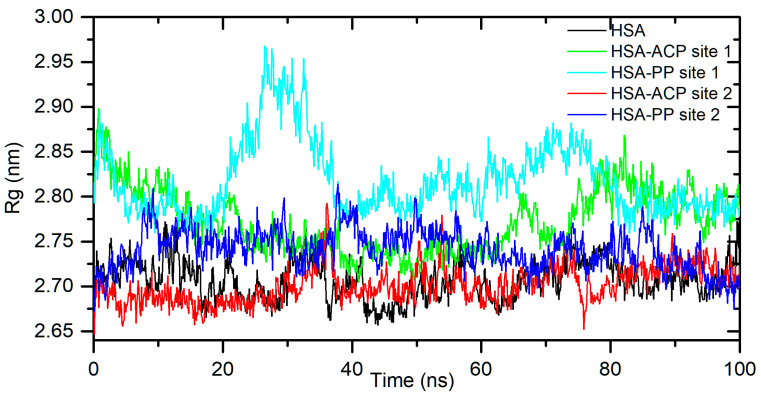
Rg plots of free HSA and for complex with ACP and PP docked at site 1 and site 2, during 100 ns MD simulation.

**Figure 6 ijms-25-05398-f006:**
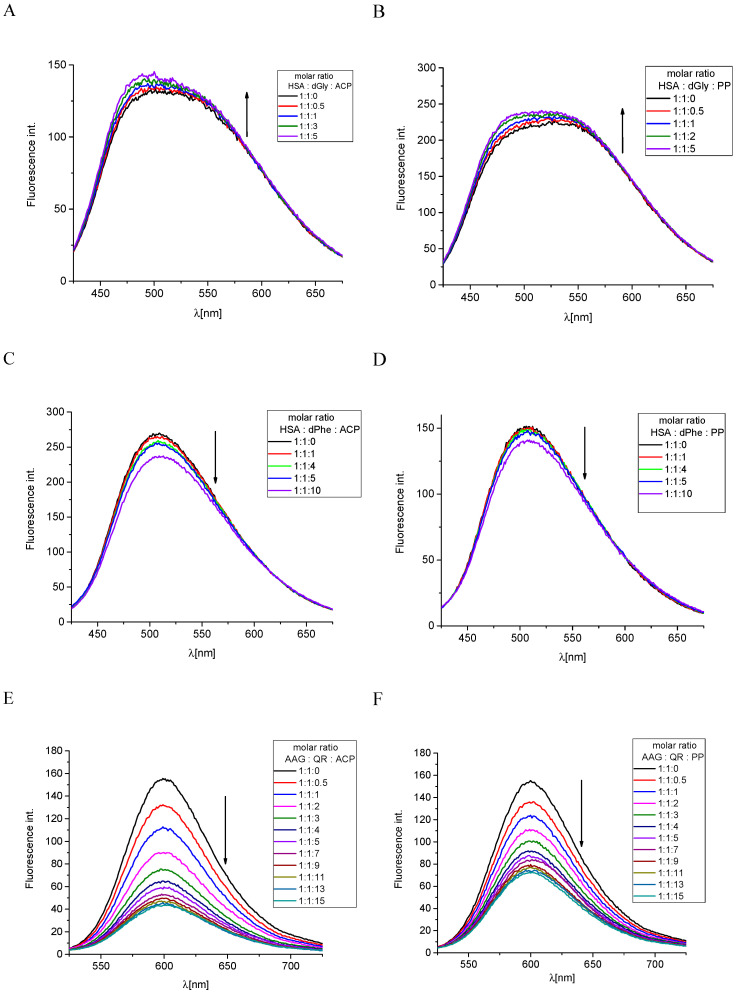
The fluorescence spectra of (**A**) HSA/dGly complex titrated with the studied compound ACP, (**B**) HSA/dGly complex titrated with the studied compound PP, (**C**) HSA/dPhe complex titrated with the studied compound ACP, (**D**) HSA/dPhe complex titrated with the studied compound PP, (**E**) AAG/QR complex titrated with the studied compound ACP, and (**F**) AAG/QR complex titrated with the studied compound PP.

**Figure 7 ijms-25-05398-f007:**
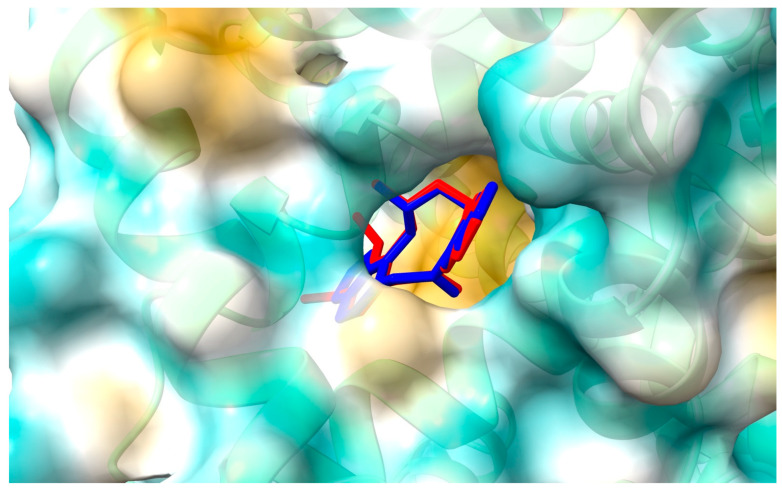
Location of allocryptopine (red) and protopine (blue) at the HSA binding site 2.

**Figure 8 ijms-25-05398-f008:**
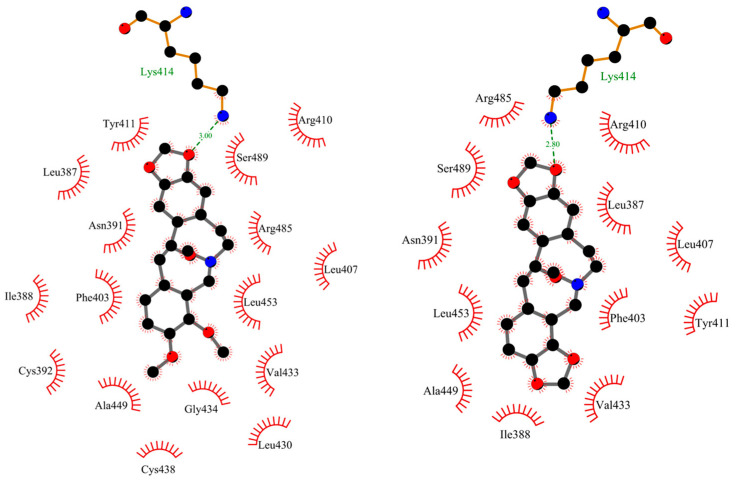
The 2D plot of interactions between allocryptopine (**left**) and protopine (**right**) with HSA at the binding site 2.

**Figure 9 ijms-25-05398-f009:**
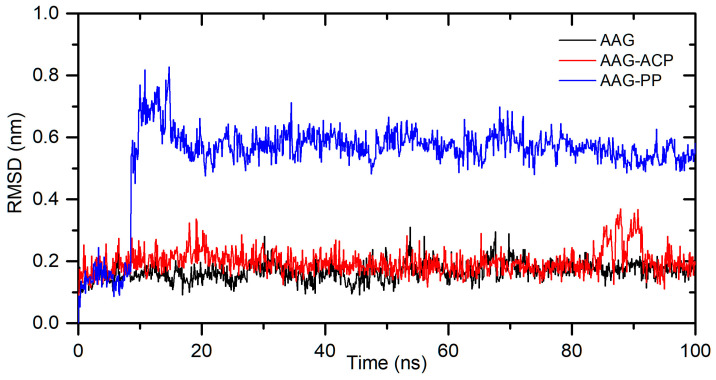
The backbone atoms’ RMSD plots during 100 ns MD simulation for AAG, AAG-ACP, and AAG-PP.

**Figure 10 ijms-25-05398-f010:**
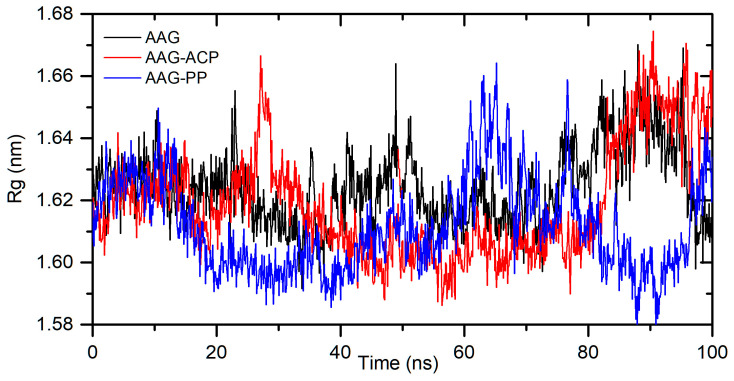
Rg plots during 100 ns MD simulation for AAG, AAG-ACP, and AAG-PP.

**Figure 11 ijms-25-05398-f011:**
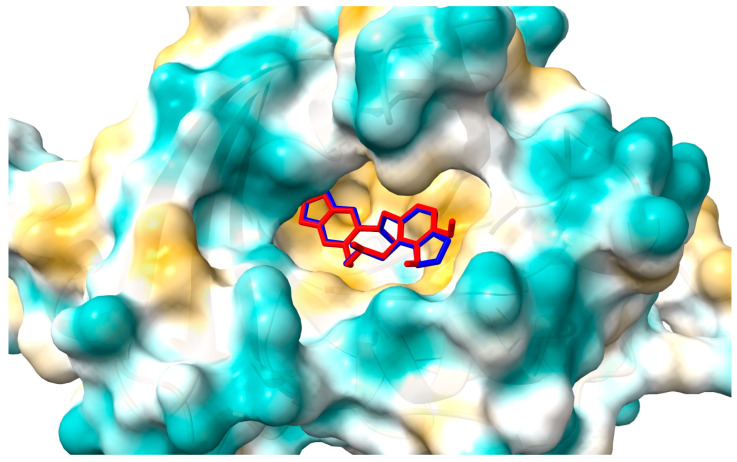
Location of allocryptopine (red) and protopine (blue) in the AAG pocket.

**Figure 12 ijms-25-05398-f012:**
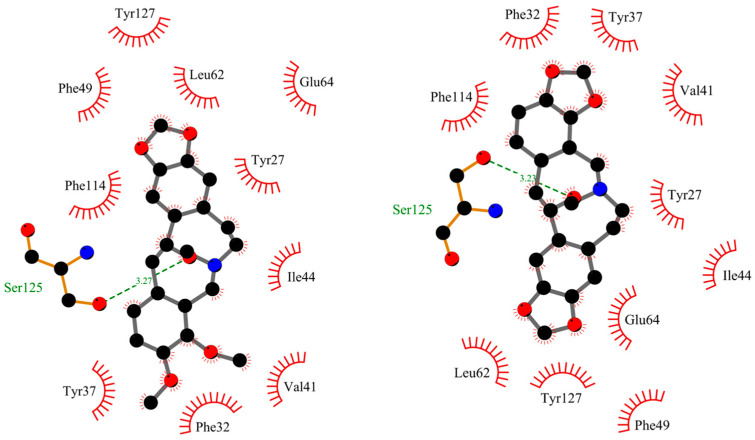
The 2D plot of interactions between allocryptopine (**left**) and protopine (**right**) with AAG.

**Figure 13 ijms-25-05398-f013:**
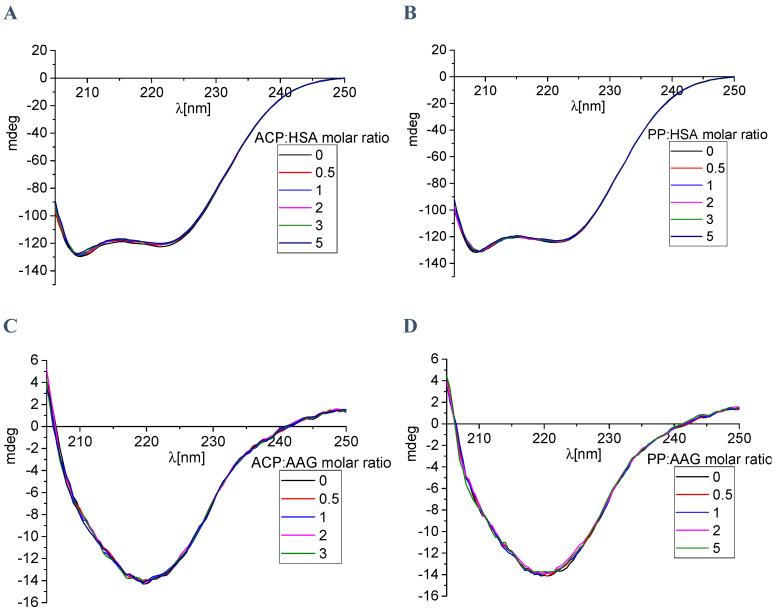
The circular dichroism spectra of: (**A**) HSA with ACP, (**B**) HAS with PP, (**C**) AAG with ACP, and (**D**) AAG with PP.

**Table 1 ijms-25-05398-t001:** The binding parameters for interaction between HSA and AAG proteins and analyzed ACP and PP alkaloids: the association constant (K_app_), the number of binding sites (n*), the standard Gibbs free energy (ΔG), and the percentage of hyperchromicity effect.

Protein/Alkaloid	K_app_·10^4^ [dm^3^mol^−1^]	n*	ΔG [Jmol^−1^]	% Chromism
**HSA/ACP**	5.02 ± 0.12	0.97	−2.68 × 10^4^	65.52
**HSA/PP**	2.13 ± 0.06	0.77	−2.47 × 10^4^	71.50
**AAG/ACP**	2.24 ± 0.04	0.75	−2.48 × 10^4^	73.50
**AAG/PP**	1.34 ± 0.0.03	0.84	−2.35 × 10^4^	81.93

**Table 2 ijms-25-05398-t002:** The binding affinity (kcal/mol), calculated from molecular docking studies, and the binding free energy (kcal/mol), calculated from molecular dynamics simulations, for HSA-ACP, HSA-PP, AAG-ACP, and AAG-PP systems.

	ACP		PP	
	Binding Affinity	Binding Free Energy	Binding Affinity	Binding Free Energy
**HSA, site IIA**	−6.3	−17.6	−6.5	−12.3
**HSA, site IIIA**	−7.7	−20.1	−7.7	−18.4
**AAG**	−8.8	−17.0	−9.8	−23.3

**Table 3 ijms-25-05398-t003:** The percentage of marker dPhe and QR replacement (PMR) in proteins HSA or AAG, respectively, by the studied compounds ACP and PP. The PMR parameter was calculated according to equation E7.

Molar Ratio HSA:dPhe:Compound	PMR Compound	
	ACP	PP
1:1:0.5	1.8%	1.3%
1:1:1	2.5%	1.5%
1:1:4	4.8%	1.6%
1:1:5	5.5%	3.2%
1:1:10	12.1%	6.7%
**Molar Ratio** **AAG:QR:Compound**		
1:1:0.5	15.4%	13.0%
1:1:1	28.1%	20.1%
1:1:2	42.6%	29.2%
1:1:3	51.6%	35.3%
1:1:4	59.0%	41.1%
1:1:5	62.2%	44.1%
1:1:7	66.0%	46.4%
1:1:9	68.0%	48.7%
1:1:11	70.1%	50.4%
1:1:13	71.6%	52.7%
1:1:15	71.9%	53.7%

**Table 4 ijms-25-05398-t004:** The percentage of the secondary structure elements in HSA in the absence and presence of ACP and PP. The values were calculated with the CD Multivariate SSE program.

HSA:Analyzed Ligand Molar Ratio	% α-Helix	% β-Sheet	% β-Turn	% Other
**ACP**				
**1:0**	64.8%	1.5%	9.7%	24.0%
**1:0.5**	64.2%	1.8%	9.7%	24.2%
**1:1**	64.1%	2.2%	9.7%	23.9%
**1:2**	63.9%	2.2%	9.7%	24.2%
**1:3**	63.9%	2.2%	9.7%	24.2%
**1:5**	63.9%	3.0%	9.6%	23.5%
**PP**				
**1:0**	65.7%	0.5%	9.7%	24.1%
**1:0.5**	65.7%	1.1%	9.6%	23.6%
**1:1**	65.6%	1.3%	9.6%	23.5%
**1:2**	65.4%	1.1%	9.6%	23.8%
**1:3**	65.7%	1.9%	9.5%	22.9%
**1:5**	65.4%	1.6%	9.6%	23.4%

**Table 5 ijms-25-05398-t005:** The percentage of the secondary structure elements in AAG in the absence and presence of ACP and PP. The values were calculated with the CD Multivariate SSE program.

AAG:Analyzed Ligand Molar Ratio	% α-Helix	% β-Sheet	% β-Turn	% Other
**ACP**				
**1:0**	20.6%	36.0%	10.7%	32.7%
**1:0.5**	20.5%	35.8%	10.8%	32.9%
**1:1**	20.6%	35.5%	10.9%	33.0%
**1:2**	20.7%	36.3%	10.7%	32.3%
**1:3**	20.7%	35.8%	10.8%	32.7%
**1:5**	20.5%	35.5%	10.8%	33.1%
**PP**				
**1:0**	20.7%	35.6%	10.8%	32.9%
**1:0.5**	20.7%	36.0%	10.8%	32.6%
**1:1**	20.7%	36.1%	10.8%	32.4%
**1:2**	20.3%	35.8%	10.9%	33.0%
**1:3**	20.4%	35.8%	10.8%	33.0%
**1:5**	20.5%	35.9%	10.8%	32.8%

## Data Availability

The data are available on request from the corresponding authors.

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
