# Peer review of "Protopine and Allocryptopine Interactions with Plasma Proteins"

_ijms, 2024, doi:10.3390/ijms25105398_

Round 1

Reviewer 1 Report

Comments and Suggestions for Authors

The article addresses an interesting topic that is widely explored by the scientific community. From my perspective, the manuscript can be published with some corrections.

 -In the introduction, the authors provided a very simplified review, with only two recent references (1-2022, 1-2024). Since the interaction of ligands with plasma proteins is a well-explored topic, the authors should expand their review and add more references to the introduction.

-The authors mention two proteins and their functions in the introduction. As this manuscript concerns the interaction of proteins with small molecules, the authors should add structural and physicochemical characteristics of these proteins and the ligands

-In section "2.1. Determination of binding constants," the authors presented experimental results without associating any errors with them. The authors should include the experimental error in the results.

-In the results section, the authors only described the obtained results but did not compare them with results reported in the literature. The authors should add a discussion of results reported in the literature.

-Regarding the methodology, the authors did not mention whether duplicates or triplicates of UV-Vis absorption, Fluorescence, and Circular Dichroism experiments were performed. The authors should clarify this and add it to the manuscript.

-In the fluorescence experiments, the authors did not mention if inner filter correction was performed. If not, the quenching of the fluorescence signal is a mere artifact. The authors should clarify this point and add it to the manuscript.

-The authors did not mention the criteria for choosing the conformation in the molecular docking calculations. The authors should include this in the manuscript and add in the supplementary material the graph of conformations obtained by interaction energy.

-The authors did not mention if duplicates of the molecular dynamics simulation were performed and whether the parameters of the trajectories were averaged. The authors should clarify this and include it in the manuscript with the RMSD and RG graphs of the independent simulations in the supplementary material.

Comments on the Quality of English Language

Minor editing of English language required

Author Response

Reviewer 1:

The article addresses an interesting topic that is widely explored by the scientific community. From my perspective, the manuscript can be published with some corrections.

Query 1. In the introduction, the authors provided a very simplified review, with only two recent references (1-2022, 1-2024). Since the interaction of ligands with plasma proteins is a well-explored topic, the authors should expand their review and add more references to the introduction.

Author’s response: Thank you for this suggestion. The Introduction section has been enhanced with a broader review of literature and an expanded discussion on the interactions of ligands with plasma proteins.

Query 2. The authors mention two proteins and their functions in the introduction. As this manuscript concerns the interaction of proteins with small molecules, the authors should add structural and physicochemical characteristics of these proteins and the ligands

Author’s response: Thank you for this suggestion as well. Structural and physicochemical characteristics of both, the proteins and the ligands have been added to the Introduction section.

Query 3. In section "2.1. Determination of binding constants," the authors presented experimental results without associating any errors with them. The authors should include the experimental error in the results.

Author’s response: Errors for binding constants have been added to the data in Table 1.

Query 4. In the results section, the authors only described the obtained results but did not compare them with results reported in the literature. The authors should add a discussion of results reported in the literature.

Author’s response: Thank you for this valuable suggestion. The Result section is supported by relevant literature references, and the Discussion section has been enriched with additional citations that thoroughly validate the scope and legitimacy of the experiments conducted.

Query 5. Regarding the methodology, the authors did not mention whether duplicates or triplicates of UV-Vis absorption, Fluorescence, and Circular Dichroism experiments were performed. The authors should clarify this and add it to the manuscript.

Authors’ response: Thank you for bringing this oversight to our attention. We have completed the missing information in the text about UV-Vis and fluorescence experiments:

Lines 361-362 and 400-401: “Each spectrum was measured using 3 accumulations, and the final result is the average of all measurements.”

For CD measurements, information about the amount of accumulation was included in the text, but we have specified it for clarity:

Lines 410-411: “Three accumulations were used during all measurements, and the final result is the average of all measured spectra.”

Query 6. In the fluorescence experiments, the authors did not mention if inner filter correction was performed. If not, the quenching of the fluorescence signal is a mere artifact. The authors should clarify this point and add it to the manuscript.

Authors’ response: Thank you very much for drawing the Reviewer's attention to this fact. Indeed, internal filter correction is the most common method employed during fluorescence measurements. However, in the case of our measurements, it was not necessary to apply this correction. No constant values were determined based on these measurements. The percentage of marker replacement was the sole variable considered. Given that both the protein and the tested compounds lack any UV-Vis bands at the excitation and emission wavelengths of the markers, it was unnecessary to consider the aforementioned correction.

Query 7. The authors did not mention the criteria for choosing the conformation in the molecular docking calculations. The authors should include this in the manuscript and add in the supplementary material the graph of conformations obtained by interaction energy.

Authors’ response: We thank the Reviewer for these two important comments. We chose the best model based on the energy function analysis. Of course, we know that this is a statistical calculation and an important factor in choosing the best conformer is also the number of representatives for each energy score (clustering analysis). There would be no problem to include an additional graph showing this relationship if we performed molecular docking using the AutoDock script. Unfortunately, we used AutoDock Vina script. In the output file, there is no information on how many conformers there are for each scoring (cluster). More, there is not even information on how many conformers the energy score was calculated. AutoDock Vina's calculation algorithm is slightly different from AutoDock. The docking calculation consists of many independent runs, starting from random conformations. Each of these runs consists of many sequential steps. Each step involves a random perturbation of the conformation followed by a local optimization (using the Broyden-Fletcher-Goldfarb-Shanno algorithm) and a selection in which the step is either accepted or not. Each local optimization involves many evaluations of the scoring function. The number of evaluations in a local optimization is guided by convergence and other criteria. The number of steps in a run is determined heuristically, depending on the size and flexibility of the ligand and the flexible side chains. Unlike in AutoDock, in AutoDock Vina, each run can produce several results. Promising intermediate results are remembered. These are merged, refined, clustered, and sorted automatically to produce the final result. The output file contains several modes (the number specified in the configuration file) with different energy scoring and RMSD values relative to the best mode. Unfortunately, there is no information on the representation per cluster.

Query 8. The authors did not mention if duplicates of the molecular dynamics simulation were performed and whether the parameters of the trajectories were averaged. The authors should clarify this and include it in the manuscript with the RMSD and RG graphs of the independent simulations in the supplementary material.

Authors’ response: The molecular dynamics simulation was performed once for each system with the parameters described in the experimental section. We did not repeat this simulation.

All modifications made to the text are marked with track changes.

Additionally, the title of the manuscript was changed and the structures of protopine and allocryptopine were included, in response to the second Reviewer's comment.

Reviewer 2 Report

Comments and Suggestions for Authors

 In the submitted manuscript, Marciniak et al. investigated the protein complexes of two isoquinolone alkaloids: allocryptopine and protopine. The methods employed were traditional yet effective, encompassing spectroscopy and molecular modeling techniques. While the applied methods lack innovation, the published data contain sufficient novel information for publication.

 My suggestions:

1. Change the title. Not allocryptopine and protopine are the most important isoquinolones.

2. Incorporate the molecular structures of the molecules into the introduction section.

3. Expand the introduction to include other characterization techniques for protein binding, citing relevant references such as chromatography and isothermal titration calorimetry (ITC).

4.  Do the authors possess information regarding the protein binding of other isoquinolones?

5. Some words/sentences are necessary that highlight the novelty of the work

6. Line 75 Im not sure that this sentence is correct. Proton pump inhibitors, like omeprazole has also fluorescence activity at 285 nm, and in the literature you can find many work regarding HSA binding of PPIs. It is true, that the results is questionable without calculation of inner filter effects. I suggest to rethink this sentence, because now it is mean that your investigated compounds are not important.

7. Line 97 Why E0? It should be (1)

8.  Warfarin and ibuprofen (or thyroxine) are used widely as a site marker.

Author Response

Reviewer 2

In the submitted manuscript, Marciniak et al. investigated the protein complexes of two isoquinolone alkaloids: allocryptopine and protopine. The methods employed were traditional yet effective, encompassing spectroscopy and molecular modeling techniques. While the applied methods lack innovation, the published data contain sufficient novel information for publication.

My suggestions:

Query 1. Change the title. Not allocryptopine and protopine are the most important isoquinolones.

Authors’ response: The title has been changed into ‘Protopine and Allocryptopine Interaction with Plasma Proteins’

Query 2. Incorporate the molecular structures of the molecules into the introduction section.

Authors’ response: The molecular structures of protopine and allocryptopine have been incorporated I the Introduction section of the manuscript.

Query 3. Expand the introduction to include other characterization techniques for protein binding, citing relevant references such as chromatography and isothermal titration calorimetry (ITC).

Authors’ response: Characterization techniques for protein binding, citing relevant references such as chromatography and isothermal titration calorimetry (ITC) has been introduced to the Introduction section.

Query 4.  Do the authors possess information regarding the protein binding of other isoquinolones?

Authors’ response: Thank you for this suggestion. The appropriate information on the protein binding of other isoquinoline alkaloids has been introduced to the Introduction section.

Query 5. Some words/sentences are necessary that highlight the novelty of the work

Authors’ response: Thank you for this comment. The Conclusions section has been enriched with the statements highlighting the novelty of the work.

Query 6. Line 75 Im not sure that this sentence is correct. Proton pump inhibitors, like omeprazole has also fluorescence activity at 285 nm, and in the literature you can find many work regarding HSA binding of PPIs. It is true, that the results is questionable without calculation of inner filter effects. I suggest to rethink this sentence, because now it is mean that your investigated compounds are not important.

Authors’ response: We would like to express our gratitude to the Reviewer for valuable contribution. We would like to clarify some information here. For example, in the following work: https://doi.org/10.1016/j.jlumin.2012.03.01, the authors do not mention anything about the 285 nm excitation for omeprazole and other similar compounds. The described excitation concerns a protein molecule. It can therefore be concluded that even if the tested drugs show fluorescence under these conditions, it has a negligible impact on the effect on the protein. In the case of protopine and allocryptopine, even the use of inner filter correction does not significantly alter the conduct of the experiment. The fluorescence efficiency of these compounds is greater than that of protein molecules. We attempted to perform this measurement, but were unable to cope with the overlap of both effects. Consequently, we posit that the assertion we have made is indeed accurate in the case of the compounds under consideration.

Query 7. Line 97 Why E0? It should be (1)

Authors’ response: We would like to thank the Reviewer for attention. We acknowledge this observation that the equation numbering in the article is incorrect. The numbering of reactions and equations has been corrected.

Query 8. Warfarin and ibuprofen (or thyroxine) are used widely as a site marker.

Authors’ response: Indeed, the compounds mentioned by the Reviewer are commonly used as markers for HSA. However, the use of these compounds necessitates a slight modification in the methodology employed for measurement. In this instance, the values of constants calculated for systems with and without a marker are compared. Subsequently, an excitation line must be employed for the protein molecule, which was not feasible in the case of the two compounds under consideration. Consequently, the use of dansyl amino acids was deemed appropriate.

Round 2

Reviewer 2 Report

Comments and Suggestions for Authors

The manuscript is accaptable for publication in the present form.